

# Residual matrix product state for machine learning

**Ye-Ming Meng[1,2], Jing Zhang[3], Peng Zhang[3], Chao Gao[2,4]⋆ and Shi-Ju Ran[5]†**

**1** State Key Laboratory of Precision Spectroscopy, East China Normal University,
Shanghai 200062, China
**2** Department of Physics, Zhejiang Normal University, Jinhua, 321004, China
**3** School of Computer Science and Technology, Tianjin University, Tianjin, China
**4** Key Laboratory of Optical Information Detection and Display Technology of Zhejiang,
Zhejiang Normal University, Jinhua, 321004, China
**5** Department of Physics, Capital Normal University, Beijing 100048, China

⋆ gaochao@zjnu.edu.cn , † sjran@cnu.edu.cn

## Abstract

Tensor network, which originates from quantum physics, is emerging as an efficient tool for classical and quantum machine learning. Nevertheless, there still exists a considerable accuracy gap between tensor network and the sophisticated neural network models for classical machine learning. In this work, we combine the ideas of matrix product state (MPS), the simplest tensor network structure, and residual neural network and propose the residual matrix product state (ResMPS). The ResMPS can be treated as a network where its layers map the "hidden" features to the outputs (e.g., classifications), and the variational parameters of the layers are the functions of the features of the samples (e.g., pixels of images). This is different from neural network, where the layers map feed-forwardly the features to the output. The ResMPS can equip with the non-linear activations and dropout layers, and outperforms the state-of-the-art tensor network models in terms of efficiency, stability, and expression power. Besides, ResMPS is interpretable from the perspective of polynomial expansion, where the factorization and exponential machines naturally emerge. Our work contributes to connecting and hybridizing neural and tensor networks, which is crucial to further enhance our understanding of the working mechanisms and improve the performance of both models.



# 1 Introduction

The tensor network (TN), as a mathematical model widely used to describe quantum many-body states [1–4], has been successfully applied to machine learning (ML). For instance, TN is used in supervised and unsupervised image classification, natural language processing, etc. [5–11]. Several recent works also demonstrate TN's ability of establishing the connection between physics and artificial intelligence [12,13]. Nevertheless, despite the high interpretability of TN [14–16], there still exists a considerable performance gap between TN and neural network (NN) [7,17].

In the context of quantum physics, TN is used to represent linear ansatz. While in machine learning, TN realizes a non-linear map from the features to the outputs, where there exists a local kernel function [5] that maps the features of the samples to the quantum states in Hilbert space. It is still an open issue to determine whether the NN techniques can enhance TN performance. Several recent works have explored different ways to combine TN and NN, which includes adopting the convolutional neural network (CNN) as a feature extractor in TN [7,17,18]; compressing the linear layers of deep NN by matrix product operators [19]; and implementing the convolutional operations using TN [20], etc. These attempts further motivate us to investigate the possible hybridization of TN and NN.

In this work, we incorporate the information highways (also known as shortcuts) [21,22], non-linear activations, and dropout [23] into TN (MPS in specific), and propose Residual MPS (ResMPS in short). The essential underlying idea of ResMPS is a delicate way of inputting data such that the variational parameters of the network layers are the functions of the data features. Such idea is inspired by the traditional feed-forward neural network (FNN), while in FNN, the data is input only in the initial step.

We provide two specific examples of ResMPS dubbed as simple and activated ResMPS. The simple version (sResMPS in short) is a multi-linear model that can exactly be written into a standard MPS, and the activated version (aResMPS in short) is a non-linear model equipped with NN layers. The results on Fashion-MNIST show that the simple ResMPS achieves the same accuracy as MPS while its parameter complexity is half of the MPS. For the activated ResMPS,

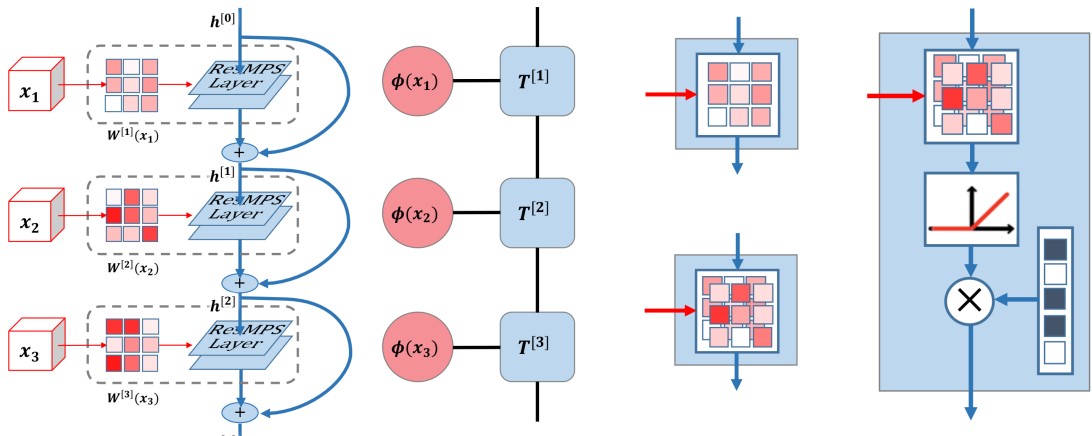

Figure 1: Illustrations of a typical ResMPS compared with a standard MPS. (a) An illustration of ResMPS containing a three-layer FNN in which the variational parameters are functions of the features, **x**. (b) An illustration of a three-tensor MPS, which is contracted with the feature vectors (see Eq. (12)). (c) An illustration of sResMPS, which is only parameterized by a single channel weight matrix. (d) An illustration of ResMPS, which is equivalent to the standard MPS. (e) An illustration of aResMPS, where the hidden feature will pass through a two-channel linear layer, ReLU activation, and dropout layer in sequence.

we find a significant enhancement of efficiency and accuracy by introducing the non-linear activations and the dropout layers on the residual terms.

Furthermore, we determine the model interpretability of sResMPS by polynomial expression. The truncated model achieves a high level of accuracy while keeping only a few low-order terms of sResMPS. Surprisingly, the factorization [24] and exponential machines [25] have naturally emerged in this expansion scheme. ResMPS shows the underlying connections between TN and NN for ML and can shed light on novel possibilities and flexibility for developing powerful ML models beyond NN or TN.

## 2 Residual matrix product state

### 2.1 Definition of residual matrix product state

The traditional FNN, including the residual neural network, consists of multiple trainable layers [26]. For instance, in supervised learning, FNN maps the input sample **x** to the output $l$, e.g., sample classification. Typically, its layer form writes

$$\mathbf{h}^{[n]} = \sigma\left(F^{[n]}\left(\mathbf{h}^{[n-1]}; \mathbf{W}^{[n]}\right) + \mathbf{b}^{[n]}\right),\tag{1}$$

where $\mathbf{h}^{[n-1]}$ denotes the hidden variables that are input to the $n$-th layer with $\mathbf{h}^{[0]} = \mathbf{x}$, $F^{[n]}$ denotes the mapping of the $n$-th layer (e.g., fully connected, convolution, or pooling layer). Each layer may consist some variational parameters $\mathbf{W}^{[n]}$ (weights) and $\mathbf{b}$ (bias). Furthermore, $\sigma$ denotes the activation function.

Inspired by the matrix product state [27, 28] and residual neural network [21, 22], here we propose a novel machine learning architecture dubbed as residual matrix product state (ResMPS). Different from FNN (see, Eq. (1)), ResMPS does not explicitly map the features with a feed-forward network. Instead, it uses the features to parameterize FNN variational

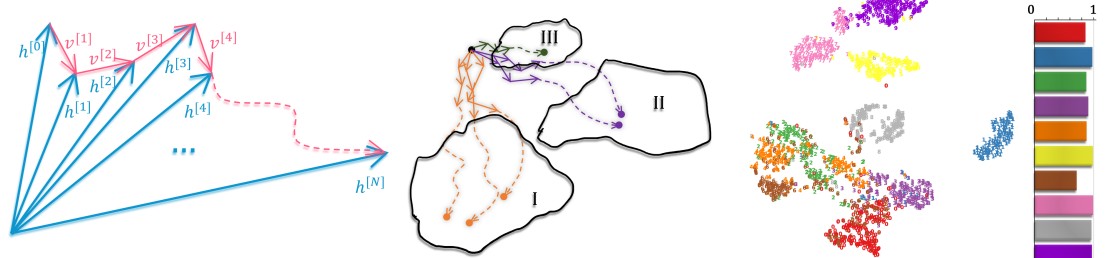

Figure 2: Encoding process of the ResMPS. (a) An illustration of a high-dimensional path of one sample. Blue arrows represent hidden features between different layers. Red arrows represent shift-vectors contributed by the residual part. (b) An illustration of the aggregation behavior of samples. The same color denotes samples belonging to the same class. (c) The two-dimensional data distribution generated by t-SNE on the endpoint dimension reduction of (b), the data points come from the Fashion-MNIST data set, and the corresponding accuracy is on the right. Note we reduce the feature dimension to 2 for illustration, which is far less than the original dimension of the hidden and weakened separations of the samples from different classes.

parameters. This enables the FNN to map the hidden features to the expected outputs (see, Fig. 1). In the ResMPS, the mapping of one layer is

$$\mathbf{h}^{[n]} = \mathbf{h}^{[n-1]} + v^{[n]}\left(\mathbf{h}^{[n-1]}; \mathbf{W}^{[n]}(x_n), \mathbf{b}^{[n]}\right), \tag{2}$$

where the weights $\mathbf{W}^{[n]}$ of the $n$-th layer are parameterized by the $n$-th feature $x_n$, $\mathbf{h}^{[0]}$ is initialized by ones for simplicity, and $v^{[n]}$ denotes the map in the $n$-th layer. Therefore, the depth of ResMPS depends on the input size. Similar to the FNN [Eq. (1)], in this work we consider $v^{[n]}$ as

$$v^{[n]} = \sigma\left(L^{[n]}\left(\mathbf{h}^{[n-1]}; \mathbf{W}^{[n]}(x_n)\right) + \mathbf{b}^{[n]}\right), \tag{3}$$

where $L^{[n]}$ is a linear map, and $\sigma$ is the activation. Similar to ResNet, the output of one layer is the addition of the output of $v^{[n]}$, and the input includes the hidden features. This is to form a shortcut of the information flow, which can avoid the vanishing/explosion of the gradients. We further note that one obtains a standard FNN by adopting $\mathbf{h}^{[0]} = \mathbf{x}$ and removing the dependence of $\mathbf{W}$ on $\mathbf{x}$.

The dimension of the hidden variable is different from the dimension of final outputs $f^{(l)}(\mathbf{x})$, where $f^{(l)}$ represents the overall mapping of the network and $l$ the classification label. Therefore, one additional linear layer without bias and activation is added to squeeze the final hidden $\mathbf{h}^{[N]}$ into the output $f^{(l)}$, i.e. $f^{(l)} = \sum_i L_{li} h_i^{[N]}$. The linear map can be flexibly replaced by other map as long as its output dimension matches the dimension of the label.

## 2.2 The working mechanism of ResMPS

We illustrate the path of the hidden state $h^{[i]}$ of ResMPS in the high-dimensional vector space (as shown in Fig.2a). Each layer of the ResMPS updates the state $h^{[i]}$ once to make it one step forward with shift-vector $v^{[i+1]} = h^{[i+1]} - h^{[i]}$. After passing through all layers, all shift-vectors are connected into a continuous path, namely $\sum_{i=1}^N v^{[i]}$. For the same ResMPS, different features of the samples share the same initial point (i.e., $h^{[0]}$). Since the parameter $W$ of shift-vector $v$ is a function of feature $x$, the path encodes the information of samples. Besides, Similar samples have close paths in the vector space (as shown in Fig.2b). After training convergence, samples of the same category will eventually gather together.

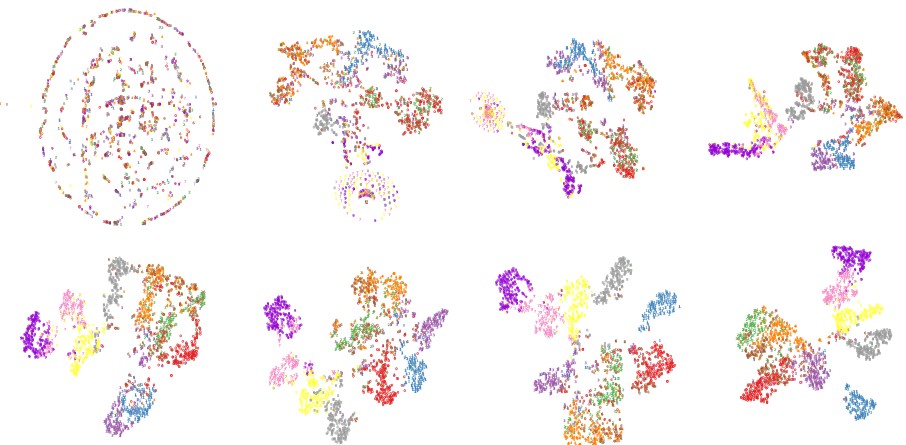

Figure 3: Intermediate hidden features $h_i^{[n]}(1 \leq n \leq 784)$ visualized by t-SNE. It is shown that samples of the same classes gradually enter the same regions. This characterizes the encoding process of ResMPS.

In order to show the behavior of paths of the hidden variables in the high-dimensional space, we use the Fashion-MNIST dataset to train sResMPS, and use the t-SNE algorithm [29, 30] to embed the endpoints of the ResMPS to a two-dimensional plane after the network converges. Note that before we apply t-SNE for dimensionality reduction, the original virtual feature has 100 components. Fig. 2c illustrates the visualization of final hidden features $h_i^{[N]}$ in the two-dimensional space. Samples with better classification accuracy are relatively separated, while those with poor classification accuracy overlap with other classifications. We also demonstrate t-SNE results of intermediate hidden features in Fig. 3 to reveal more details. It shows that the initial hidden features are uniformly distributed, and then gradually separated during the encoding progresses.

## 2.3 The Architecture of ResMPS

In the following, we examine two instances of ResMPS, called simple ResMPS (sResMPS, see Fig.1c) and activated ResMPS (aResMPS, see Fig.1f). The sResMPS is a multi-linear model that is equivalent to MPS. It achieves the same accuracy with only half of the parameter complexity of the MPS. The aResMPS is a generalized version of sResMPS, in which the generalization efficiency is enhanced by introducing non-linear activation functions and dropout in the FNN part. The map of one layer in the sResMPS is written as

$$h_j^{[n]} = h_j^{[n-1]} + \sum_i x_n W_{ij}^{[n]} h_i^{[n-1]} . \tag{4}$$

The weights of the layers in the FNN are linearly dependent on the features **x**. The bias terms are also disabled in this example.

sResMPS is equivalent to a restricted version of MPS and can achieve identical performance with only a half parameter complexity of standard MPS. See Sec. 2.4.2 for details.

It is seen that MPS has a remarkable representation power. The training error is less than 1% [31]. However, a gap between the training and testing accuracy suggests an over-fitting issue. To address the over-fitting issue, we propose the aResMPS by incorporating the non-linear activation functions and dropout. This modification also enhances the generalization power [32]. The map of each layer in the FNN of the aResMPS is more-or-less a fully-connected layer with a shortcut, which reads

$$\mathbf{h}^{[n]} = \mathbf{h}^{[n-1]} + \sigma \left( L^{[n]}(\mathbf{h}^{[n-1]}) + \mathbf{b}^{[n]} \right) , \tag{5}$$

where $\sigma$ is an activation function. The map $L^{[n]}$ relies on the feature $x_n$ in a non-linear fashion

$$L^{[n]}(\mathbf{h}^{[n-1]})_j = \sum_{c=1,2}\left[\xi^{[c]}(x_n)\sum_i W_{ij}^{[n,c]}h_i^{[n-1]}\right]. \tag{6}$$

The architecture of ResMPS is flexible, due to the choice of $\xi^{[c]}(x_n)$ and the number of channels dim($c$). Here we choose $\xi^{[1]}(x_n) = x_n$ and $\xi^{[2]}(x_n) = 1 - x_n$. We introduce $\xi^{[c]}$ to enhance the non-linearity of the aResMPS. It is worth mentioning that even sResMPS represents a non-linear map on the features $\mathbf{x}$ (but a linear map on the hidden features).

For the aResMPS, the map on either the features or the hidden features is non-linear. Indeed, the FNN embedded inside the aResMPS can be replaced by any NN. Here, we choose a standard fully-connected network with two channels labeled by $c$.

Throughout this paper, we choose the ReLU activation function that can screen the negative inputs [33, 34]. Due to its piecewise linear characteristics, the gradient can directly pass through without attenuation or enhancement. Therefore, the ReLU function is suitable for enhancing the non-linearity of the deep networks, which can improve its expression ability and avoid the vanishing/explosion of the gradient. Furthermore, we use a dropout layer combined with the residual structure to improve the generalization ability of ResMPS. The dropout layer effectively creates an ensemble of networks while avoiding the co-adaptation of intermediate variables [23, 35, 36]. We impose dropout on the residual terms, i.e., $\mathbf{h}^{[n]} = \mathbf{h}^{[n-1]} + \text{dropout}(\sigma(\cdots))$.

If we discard the activation and the dropout layers of aResMPS (see Fig.1e), we will get a standard two-channel MPS. For a standard MPS with physical bond dimension $d = 2$, the map given by a local-tensor construction is [31]

$$h_j^{[n]} = \sum_{c=1,2}\left[\xi^{[c]}(x_n)\sum_i T_{ij}^{[n,c]}h_i^{[n-1]}\right]. \tag{7}$$

If we introduce transformation $T_{ij}^{[n,c]} = W_{ij}^{[n,c]} + \delta_{ij}$, we can simply get $h_j^{[n]} = h_j^{[n-1]}\left(\sum_{c=1,2}\xi^{[c]}(x_n)\right) + \sum_{c=1,2}\left[\xi^{[c]}(x_n)\sum_i W_{ij}^{[n,c]}h_i^{[n-1]}\right]$. By taking feature map with norm-1 normalization [31], i.e. $\sum_{c=1,2}\xi^{[c]}(x_n) = 1$, we get a ResMPS with map

$$h_j^{[n]} = h_j^{[n-1]} + \sum_{c=1,2}\left[\xi^{[c]}(x_n)\sum_i W_{ij}^{[n,c]}h_i^{[n-1]}\right]. \tag{8}$$

## 2.4 Benchmarking results

### 2.4.1 Classification accuracy

For the MNIST [39] and Fashion-MNIST [40] datasets, Table 1 shows the accuracy of the sResMPS and aResMPS, compared with several established NN [38] and TN models [7, 9, 15, 17, 31, 37]. The MPS and ResMPS models represent a high level of representation power, as indicated by their high training accuracy. The aResMPS also surpasses the probabilistically interpretable Bayesian [15] and other TN models, including the two-dimensional TN known as projected-entangled pair state (PEPS) [17]. It also achieves a (slightly) better accuracy than that of CNN-PEPS model, in which CNN is adopted as the feature extractor. This accuracy surpasses the CNN without the stacking architecture, such as AlexNet [38]. The aResMPS still does not overperform the ResNet which is formed by stacking multiple convolution layers.

Table 1: Experimental results on MNIST and Fashion-MNIST dataset. The first 6 models are pure TN architectures, which means they are multi-linear, and no neural structures like pooling, activation and convolutional are introduced. AlexNet, ResNet, and CNN-PEPS are NN or TN-NN hybrid models. For aResMPS, we use ReLU as activation, and the dropout probability is set to be 0.6.

| Model | MNIST train | MNIST test | Fashion-MNIST train | Fashion-MNIST test |
|---|---|---|---|---|
| MPS machine [31] | 1.0000 | 0.9880 | 0.9988 | 0.8970 |
| Unitary tree TN [9] | 0.98 | 0.95 | - | - |
| Tree curtain model [37] | - | - | 0.9538 | 0.8897 |
| Bayesian TN [15] | - | - | 0.8950 | 0.8692 |
| EPS-SBS [7] | - | 0.9885 | - | 0.886 |
| PEPS [17] | - | - | - | 0.883 |
| CNN-PEPS [17] | - | - | - | 0.912 |
| AlexNet [38] | - | - | - | 0.8882 |
| ResNet [38] | - | - | - | 0.9339 |
| sResMPS | 1.0000 | 0.9873 | 0.9987 | 0.8909 |
| aResMPS | 1.0000 | 0.9907 | 0.9999 | 0.9142 |

### 2.4.2 Redundancy of regular MPS

To see the equivalence to the standard MPS and sResMPS mentioned in Sec.2.3, let us introduce the third-order tensors $\mathbf{T}^{[n]}$ satisfying

$$T_{1,:,:}^{[n]} = \mathbf{I}, \quad T_{2,:,:}^{[n]} = \mathbf{W}^{[n]}. \tag{9}$$

The feature vectors $\phi(x_n)$ are obtained by the feature map as $\phi(x_n) = (1, x_n)$, similar to Refs. [5,9,31]. Summing their joint index gives a linear mapping represented by a matrix

$$A_{a_{n-1},a_n}^{[n]} = \sum_{p_n} T_{p_n,a_{n-1},a_n}^{[n]} \phi(x_{n-1})_{p_n} = \delta_{a_{n-1},a_n} + x_n W_{a_{n-1},a_n}^{[n]}. \tag{10}$$

Applying this mapping to $h^{[n-1]}$, one gets

$$h^{[n]} = \sum_{a_{n-1}} h_{a_{n-1}}^{[n-1]} A_{a_{n-1},a_n}^{[n]} = h_{a_n}^{[n-1]} + \sum_{a_{n-1}} h_{a_{n-1}}^{[n-1]} x_n W_{a_{n-1},a_n}^{[n]}, \tag{11}$$

this form is exactly the definition of sResMPS, see Eq. (4). Therefore, the sResMPS is equivalent to the standard MPS formed by the following tensors

$$\mathcal{T} = \sum_{\{a_n\}} \prod_n T_{p_n a_n a_{n+1}}^{[n]}, \tag{12}$$

as its tensor-train cores [41] [Fig. 1 (b)]. The numbers of the input and output hidden features for different layers provide the two virtual bond dimensions of the MPS, i.e., $\{\dim(a_n)\}$. In this work, we fix $\dim(a_n) = \chi$, $\forall n$. The physical dimension of the MPS should also match the dimension of the feature vector, i.e. $\dim(\phi(x_n)) = \dim(p_n)$.

For $\dim(p_n) = 2$, the number of variational parameters in sResMPS is $\sim O(N\chi^2)$ where $N$ is total number of features. This is only half of that in the MPS which is $\sim O(2N\chi^2)$. Our numerical simulations show that the accuracy of both models is almost the same. See the training and testing accuracy versus epochs on Fashion-MNIST dataset [40] in Fig. 4 (a) with $\chi = 40$. This is because one of the two channels of each tensor in the MPS is much less "activated". The inset of Fig. 4 (a) shows the average norm of the two channels of different tensors

$$q_p^{[n]} = \frac{1}{\chi^2} \sum_{j=1}^{\chi} \sum_{k=1}^{\chi} \left| T_{pjk}^{[n]} - \delta_{jk} \right|, \tag{13}$$

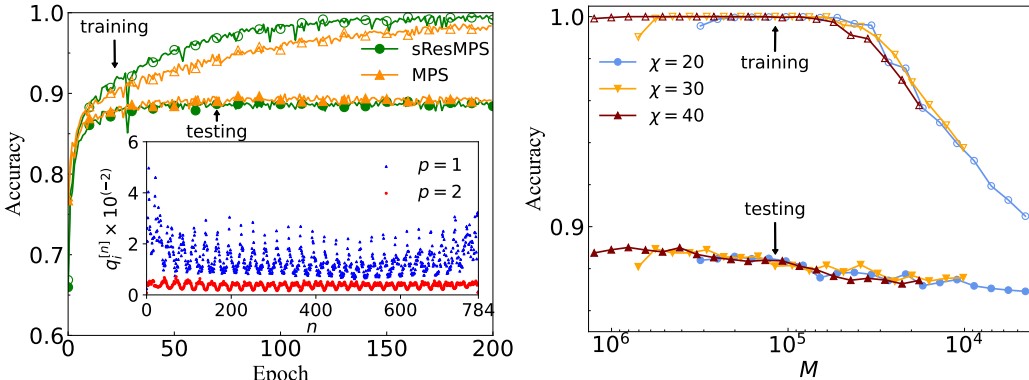

Figure 4: Numerical results of the simple ResMPS. (a) Training and testing accuracy of sResMPS (without dropout) and MPS versus epochs on the Fashion-MNIST dataset. The inset shows the average norm [Eq. (13)] of the two channels in the MPS for different tensors $n$. (b) Training and testing accuracy of the sResMPS versus the total number of unmasked weights in the sResMPS. The left end of each curve corresponds to the un-pruned result. It is also seen that the first few steps of pruning improve the accuracy. Note the total number of parameters of sResMPS with $\chi = 20, 30$, and $40$ equals to about $3 \times 10^5$, $7 \times 10^5$ and $13 \times 10^5$, respectively.

with $p = 1, 2$ representing the channels. The main contribution to the output is from the second channel. Therefore, one channel is sufficient to propagate the information to the output.

### 2.4.3 Representation power of ResMPS

In physics, the virtual bond dimension, $\chi$, characterizes the representation power of the MPS. This is because it determines the total number of variational parameters and the upper bound of the entanglement entropy the MPS can carry [1]. This may not be the case for machine learning. We show this by adding masks on the variational parameters, i.e., pruning [14]. Each parameter is multiplied by a factor that is either zero or one. The parameters multiplied by zeros are masked. To mask a certain number of parameters, we choose to mask those with relatively small absolute values. We then optimize the unmasked parameters after the masks take effect.

Fig. 4 (b) shows the accuracy values versus the number of unmasked parameters $M$. For different virtual bond dimensions, $\chi = 20, 30$, and $40$, the results are similar if the number of the unmasked parameters are the same. This suggests that the parameter which charac-terizes the representation and generalization power, is in fact, M (not $\chi$). For a given $\chi$, it is possible to further reduce the complexity of MPS (and sResMPS) without harming the accu-racy. Our results also indicate that the sResMPS achieves its maximal representation power for $M \sim O(10^4)$ (the training accuracy $\simeq 99.98\%$).

### 2.5 Similar netowrks

The layer mapping of ResMPS is similar to recurrent neural network (RNN). However, there are two essential differences between ResMPS and RNN. Firstly, ResMPS is non-uniform, and can be generalized to the translation invariant MPS named as uniform MPS. From this point of view, RNN, in which the layers share the same variational parameters, is closer to the uniform MPS. Secondly, the RNN layers are constructed by several linear operations and non-linear activations, so networks like long short-term memory (LSTM) can be classified as an RNN. For the ResMPS, it is formed by tensor units, which are simply multi-linear mappings and can be

analyzed by the established methods such as tensor decompositions [41].

ResMPS is also different from transformer networks. Firstly, the transformer network is established on the attention mechanism, while the working mechanism of ResMPS is explained in Sec. 2.2. Secondly, A general transformer network consists of an encoder and a decoder. The encoder transforms a data sequence into a vector, while the decoder transforms the vector into another sequence. In contrast, ResMPS, as well as MPS-based networks only output a vector. Moreover, ResMPS does not need positional encoding, which is the way that sequential information enters a transformer network. Considering all these differences, we regard ResMPS as a new model different from all previous ones.

## 3 Properties of the residual structure

### 3.1 Avoiding the gradient problems by residual terms

A typical MPS architecture designed for pattern recognition contains hundreds of tensor cores. Such an architecture probably encounters gradient vanishing/exploding problems. For this reason, some existing MPS schemes apply a DMRG-like algorithm where the MPS takes the canonical form [5,6,10,42]. In these attempts, however, the accuracy is sensitive to the hidden features' dimensions (virtual bonds). Recently, an MPS algorithm was proposed based on the automatic gradient technique [31] that can achieve higher accuracy than the previous ones, while its performance is not sensitive to the virtual dimensions. To find out why such a deep network avoids the gradient problems, here we construct the tensor cores to satisfy a specific form given by Eq. (9). The identity in $T_{1,:,:}^{[n]}$ plays the role of "highway" to pass the information from the previous tensor core directly to the latter ones. The components $T_{2,:,:}^{[n]}$ represent the residual terms, which is $\ll O(1)$. The application of residual conditions implies that each layer of ResMPS can easily express identity mapping. In other words, the architecture of ResMPS satisfies the identity parameterization [21, 22, 43].

To further demonstrate the role of identity parameterization in ResMPS, we use Gaussian distributions with zero mean and standard deviation $\varepsilon$ to randomly initialize the elements of $T_{2,:,:}^{[n]}$. Fig. 5 shows the testing accuracy at the 10-th, 20-th, and 50-th epochs. For a sufficiently small $\varepsilon$, the accuracy is quickly and stably converged. However, for relatively large $\varepsilon$ (e.g., $O(10^{-1})$) as illustrated by the green region, the gradients become unstable. Consequently, the accuracy stays around 0.1 and cannot improve further by the training process. Note that this may be unstable in most cases if instead of the identity parameterization, the entire $T$ is randomly initialized.

### 3.2 Relations to polynomial expansion

The forward propagation of the sResMPS (4) is fully linear on the hidden features. Applying the maps to the initial hidden features $\mathbf{h}_0$ in sequence, we can then rewrite the output hidden features in an expansive form [Fig. 6 (b)] as

$$\mathbf{h}^{[N]} = \left(\mathbf{I} + x_N \mathbf{W}^{[N]}\right)\dots\left(\mathbf{I} + x_2 \mathbf{W}^{[2]}\right)\left(\mathbf{I} + x_1 \mathbf{W}^{[1]}\right)\mathbf{h}^{[0]} = \sum_{k=0}^{N} \mathbf{M}^{[k]}\mathbf{h}^{[0]}, \qquad (14)$$

where $N$ is the total number of features $\mathbf{x}$. The output $\mathbf{h}^{[N]}$ is the stack of $N$ terms. The zeroth term satisfies $\mathbf{M}^{[0]} = \mathbf{I}$, which is the result of the information highway from the first input hidden features to the output. The term $\mathbf{M}_1 = \sum_{\alpha=1}^{N} x_\alpha W^{[\alpha]}$ is the part in ResMPS which is

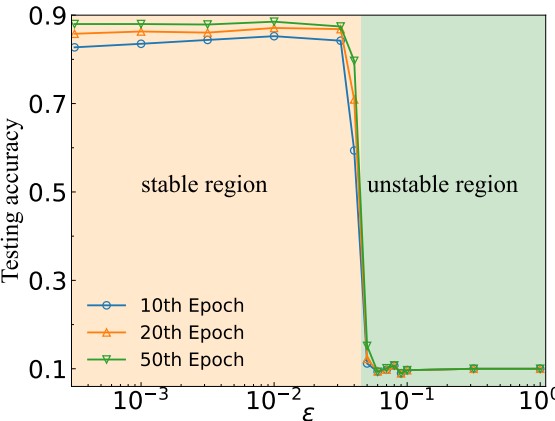

Figure 5: The testing accuracy of the sResMPS versus $\varepsilon$ on Fashion-MNIST dataset. Here $\varepsilon$ is the standard deviation of the initial residual part. We fix the number of epochs to be 10, 20, and 50. The network can be trained stably for small values of $\varepsilon$. Otherwise, the training process encounters gradient vanishing (or explosion) problems. The stable and unstable regions are illustrated by orange and green colors, respectively. Note that for the unstable region, the value of the network elements diverges. The network will give classifications randomly, thus, the accuracy tends to be 0.1.

linear on the features $\mathbf{x}$. The $k$-th term contains the $k$-th order contributions from $\mathbf{x}$, i.e.,

$$\mathbf{M}_k = \sum_{\alpha_1 \dots \alpha_k = 1}^{N} G_{\alpha_1 \dots \alpha_k} x_{\alpha_1} \dots x_{\alpha_k} \mathbf{W}^{[\alpha_1]} \dots \mathbf{W}^{[\alpha_k]}, \tag{15}$$

$$G_{a_1, a_2, \dots, a_n} = \begin{cases} 1, & a_1 > a_2 > \dots > a_n, \\ 0, & \text{otherwise}. \end{cases} \tag{16}$$

This formula is a specific form of the Exponential Machines [25]. Due to their essential similarity, the algebraic properties of Exponential Machines are also valid for sResMPS. For instance, the output feature $\mathbf{h}^{[N]}$ is a linear mapping concerning the initial hidden feature $h_0$, and a multi-linear mapping concerning the feature $\mathbf{x}$.

From the residual condition (see Eq. (9) with $|\mathbf{W}^{[n]}| \ll O(10^{-1})$), the contributions from the higher-order terms of (15) should decay exponentially with $k$. Therefore, we can define a set of lower-order effective models by retaining the first few terms. For instance, by keeping the zeroth- and first-order terms in Eq. (15), we obtain a model in which the output features are linear to both hidden and sample features. Keeping the zeroth, linear, and quadratic terms gets a model

$$\mathbf{h}^{[N](2)} = \left( \mathbf{I} + \sum_{\alpha=1}^{N} x_\alpha \mathbf{W}^{[\alpha]} + \sum_{\alpha, \beta = 1}^{N} G_{\alpha, \beta} x_\alpha x_\beta \mathbf{W}^{[\alpha]} \mathbf{W}^{[\beta]} \right) \mathbf{h}^{[0]}. \tag{17}$$

This model is similar to Factorization Machines [24] and polynomial NN [44].

Fig. 6 (a) shows the difference between the accuracy of several lower-order models and the sResMPS. This implies that the significant improvement achieved by the sResMPS has its root in a few lower-order terms, especially the linear term. As the order increases, the cost of directly computing Eq. (14) is also exponentially increased. Therefore, truncating the order of expansion is not economical. ResMPS adopts a different and efficient scheme for retaining all higher-order interactions.

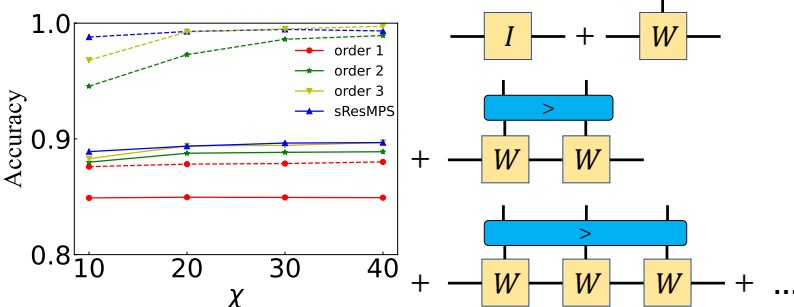

Figure 6: Polynomial expansion based on ResMPS. (a) Training (dashed) and testing (solid) accuracy versus $\chi$ by taking different orders in the expansion form; (b) The illustration of the polynomial expansion picture of the sResMPS. See Eq. (15).

## 4 Conclusion

We propose ResMPS by incorporating MPS with the information highways, non-linear activations, and dropout. In contrast to FNN, the variational parameters in ResMPS are replaced by adjustable functions. For FNN, features are input at the first layer of the network. For ResMPS, however, features are divided and input into the weight matrices of each layer, which is inherited from MPS. Furthermore, the introduction of the neural network structures results in ResMPS to have a more vital expression ability than the MPS. We also present two specific versions of ResMPS.

The first derived architecture sResMPS is a simple linear version of ResMPS. By comparing MPS' learning performance on the Fashion-MNIST dataset, we further reveal the channel redundancy of MPS. sResMPS also discards the redundant channel. Consequently, it achieves consistent accuracy while the parameter complexity is halved.

The second one is aResMPS, which is the general ResMPS equipped with activation and dropout layers. We further compare the model with several TN and NN models on the Fashion-MNIST dataset. The activation and dropout layer enhance the non-linearity and generalization ability of the model, respectively. Therefore, aResMPS surpass the state-of-the-art TN methods and AlexNet in terms of accuracy, although still inferior to ResNet formed by stacking multiple convolution layers. Going beyond present aResMPS to achieve higher accuracy, e.g., replacing the weight matrices with convolution layers, is a valuable improvement direction of ResMPS.

The perspectives of the residual network derived the polynomial expansion of ResMPS. The benefits are two-fold. Firstly, we give the condition of vanishing/explosion of the gradients of ResMPS. This helps the feature design of MPS and ResMPS algorithms with stable convergence. Secondly, it establishes the equivalence between MPS and polynomial networks such as Factorization Machines and Exponential Machines. Further numerical evidence suggests that the contribution of high-order terms is insignificant. This helps to better understand the MPS and ResMPS.

Are other NN structures (e.g., convolution and pooling layers) compatible with ResMPS? Is it possible to propose a ResMPS structure based on general NN structures (e.g., Tree TN or Projected Entangled-Pair States)? These problems are worthy of further investigation in the future.

## Acknowledgements

Y.-M.M. and C.G. are supported by National Natural Science Foundation of China (NSFC, Grant No. 1183501 and No. 12074342) and Zhejiang Provincial Natural Science Foundation of China (Grant No. LY21A040004). S.-J.R. is supported by NSFC (Grant No. 12004266 and No. 11834014), Beijing Natural Science Foundation (Grant No. 1232025), and the Academy for Multidisciplinary Studies, Capital Normal University. J.Z. and P.Z. are supported by NSFC (Grant No. 61772363).

## A   Training details

All benchmarks of this paper are implemented on MNIST and Fashion-MNIST datasets, each includes 60,000 training and 10,000 testing grayscale images with $L = 10$ labels. To fit inputs of ResMPS, we choose a specific path to reorder 2D pixels into a 1D sqeuence $\{(x_1, x_2, \ldots x_{784})|x_i \in [0,1]\}$, where 784 is the pixel number of one image. For single channel ResMPS, e.g., the sResMPS, there is no need to feature map the original data. And for double channel ResMPS, a linear feature map $(x, 1-x)$ is adapted to match the channel dimension. The predicted classification is given by the largest component

$$f(\mathbf{x}) = argmax_l f^{(l)}(\mathbf{x}), \tag{A.1}$$

where $f^{(l)}(\mathbf{x})$ is the overall mapping of ResMPS with $l = 1, \ldots L$ denotes classification. To train ResMPS, we use the Stochastic Gradient Descent (SGD) method with Adam optimizer [45] and learning rate $10^{-4}$. Data are divided into mini-batches of size 1000. We choose cross-entropy as the loss function,

$$CE = -\sum_{(\mathbf{x_i}, y_i) \in \mathcal{D}} \log \text{softmax} f^{(y_i)}(\mathbf{x}_i), \tag{A.2}$$

where

$$\text{softmax} f^{(y_i)}(\mathbf{x}_i) = \frac{e^{f^{(y_i)}(\mathbf{x}_i)}}{\sum_{l=0}^{L-1} e^{f^{(y_l)}(\mathbf{x}_i)}}. \tag{A.3}$$

Here $\mathcal{D}$ is the data set of a mini-batch, and $\mathbf{x}_i$ and $y_i$ is the $i$-th input and classification. If dropout, the corresponding probability is set to 0.6, and ReLU for the non-linear case. The whole implementation is based on the PyTorch library, and the relevant code is available on GitHub.

## B   Path independency

ResMPS was created to deal with sequential data, but general data like images are usually high-dimensional. Therefore, one needs to choose a specific path to unfold high-dimensional data. To study how path choice effects the performance, we here compare three different paths — zigzag path, Hilbert path, and random path. The first two are illustrated in Fig. (7). The zigzag path, as the most popular one, arranges data row by row; The Hilbert path preserves more neighboring information than others; As for the random path, the original position information is completely dropped. All points are collected together, shuffled completely, and then reorderd in a random way. Note that the numbers of rows and columns of the Hilbert path need to fit $2^n$ with $n$ a positive integer, so we need to extend the original image by using borders consists of zeros. For the case of MNIST and Fashion-MNIST, image size was extended from $28 \times 28$ to $32 \times 32$.

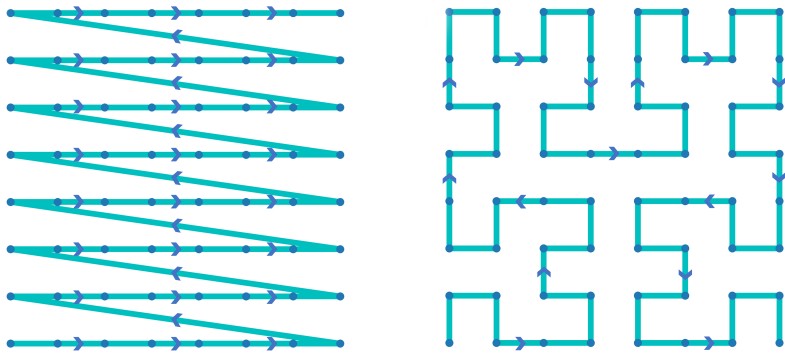

Figure 7: The zigzag path (left) and the Hilbert path (right).

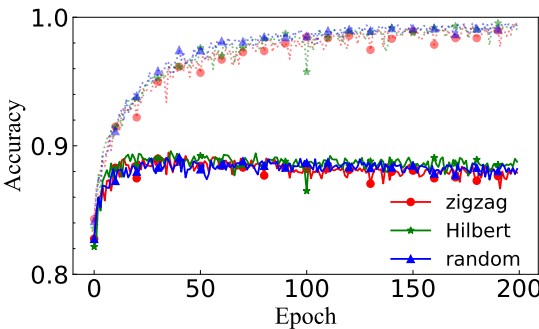

Figure 8: Convergence history of different paths on Fashion-MNIST dataset.

We run benchmarks for sResMPS without activation and dropout on the Fashion-MNIST dataset. The hidden dimensions set to be 40 and other training details are the same as appendix A. The result is shown in Fig. (8). To our surprise, there is no significant performance difference observed, even the random path still perform well.

Since the correlation decay exponentially in MPS [46], one would expect that ResMPS can not capture long-range information, and hence is path-independent. However, the previous experiment show that ResMPS is path-independent. We suppose that this inconsistency is mainly due to the introduction of residual connection, which preserve the sensitivity of gradient in the long-range, and presumably maintains the long-range correlation.

## C Additional benchmarks

aResMPS, as a double channel model equipped with non-linear activation and dropout, can achieve better performance than sResMPS. To explore the contribution of each part to the final performance, we combine different components to give more fine partitioned models. The benchmark results are shown in Tab. 2. From the experimental results, we can see that these components more or less improve the performance, and aResMPS as the most complex one achieves maximum performance.

The result of the pruning test shows the hidden dimension $\chi$ does not affect the accuracy, but this statement is only valid for sufficiently large $\chi$. If we gradually reduce the value of $\chi$, a predictable result is that the dimensions of hidden space are too small, so that different classes become hard to distinguish. We show convergence procedure of sResMPS for small $\chi$s, see Fig. 9. The results show that increasing the bond dimension indeed helps to improve the accuracy, but only for a small enough $\chi$. Once a certain threshold is exceeded, increasing $\chi$

Table 2: Benchmarks for fine partitioned models. sResMPS and aResMPS are the first and last rows respectively. Other models filled the gap between sResMPS and aResMPS.

| Model | MNIST train | MNIST test | Fashion-MNIST train | Fashion-MNIST test |
|---|---|---|---|---|
| 1-channel, -ReLU, -dropout | 1.0000 | 0.9873 | 0.9987 | 0.8909 |
| 1-channel, -ReLU, +dropout | 1.0000 | 0.9889 | 0.9618 | 0.9022 |
| 1-channel, +ReLU, -dropout | 1.0000 | 0.9864 | 1.0000 | 0.8957 |
| 1-channel, +ReLU, +dropout | 1.0000 | 0.9885 | 0.9904 | 0.9108 |
| 2-channel, -ReLU, -dropout | 1.0000 | 0.9880 | 0.9988 | 0.8970 |
| 2-channel, -ReLU, +dropout | 1.0000 | 0.9896 | 0.9899 | 0.9081 |
| 2-channel, +ReLU, -dropout | 1.0000 | 0.9873 | 1.0000 | 0.9102 |
| 2-channel, +ReLU, +dropout | 1.0000 | 0.9907 | 0.9999 | 0.9142 |

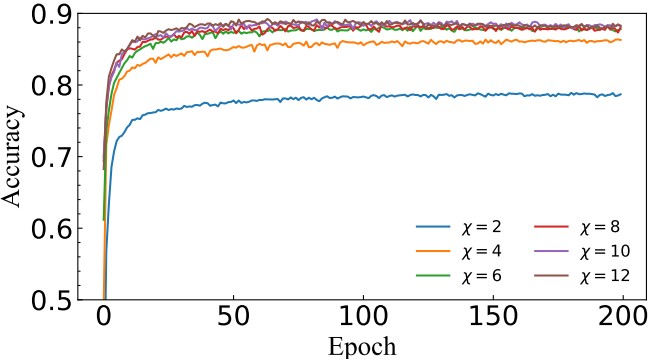

Figure 9: Testing accuracy for sResMPS on Fashion-MNIST dataset for various hidden dimension $\chi$. The performance will increase as $\chi$ increase for small $\chi$, and achieves saturation after a critical point about $\chi_c = 6$, which is the same as the target space dimension.

has no benefit to accuracy growth. And even a really small value of hidden dimension $\chi = 2$ also achieves about 80% accuracy.

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
