# Peer review of "Residual Matrix Product State for Machine Learning"

_SciPost Physics, doi:SciPost Phys. 14, 142 (2023)_

## Round 1 · Referee Report · Anonymous (Referee 2) · 2022-10-7

Strengths

See previous report.

Weaknesses

1- Conclusions drawn from the experiments on expressivity (Section 2.4.3 and Appendix C) are not supported by the data.

Report

The revised version of the manuscript is substantially improved over the previous one. In particular, the three weaknesses pointed out in my previous report were addressed and resolved.

However, the new data generated in response to my previous requested change 9) reveals another issue that touches also the claim made in the reply that "some physical concepts fails in machine learning MPS", in particular "the failure of bound dimension $\chi$ to measure the representation ability". The data in Appendix C show that the expressivity of the model is systematically improved as $\chi$ is increased. This is exactly what one expects from a parameter that controls the representation ability. Moreover, Appendix C reveals, that the performance does not increase further beyond $\chi=6$. The data shown in Fig. 4 were produced with much larger dimensions up to $\chi=40$. This corresponds to an increase of the number of parameters by two orders of magnitude. Therefore, it does not seem particularly contradictory to me that Fig. 4b) shows that one can again reduce the number of parameters by two orders of magnitude using random elimination without any effect. Clearly both ways of varying the model size are inequivalent, but I do not think that the data supports the claim that the bond dimension fails to be a measure for the representative power. In that regard, I find the discussion in Section 2.4.3 quite misleading.

Furthermore, I still think that a more machine learning oriented journal would be a better fit for the manuscript, because it has hardly any relation to physics, but this should be an editorial decision.

Overall, the manuscript will meet the acceptance criteria of SciPost Physics, if the requested changes below are addressed.

Requested changes

1- Reconsider the discussion in 2.4.3 according to my comment above. 2- Line 182: The last sentence of 2.4.1 is misleading. The data shows no evidence based on which one could say that "It seems that the ResMPS models eventually surpass ResNet...". Since it is a speculation and the same thought is repeated in the Discussion, I would suggest to remove the sentence from 2.4.1. 3- Line 48: "TN itself represents a linear map between quantum states." This formulation seems wrong. Tensor networks are not a map between quantum states. They do not map one state to another as this formulation implies. Please clarify. 4- Line 107: Should the sentence read "... the hidden variable is not equal to the dimension of the label index..."? If the two were equal, one could directly use the hidden state as output. 5- Line 122: "sResMPS" appears before it was introduced. Maybe swap the order of sections 2.2 and 2.3? 6- Line 134: Fig. 1(f) does not exist. Should be 1(e). 7- Line 364: The first sentence of this paragraph is wrong: There is no "exponential decay behavior of entanglement entropy of MPS". MPS always have a finite correlation length, i.e., correlations decay exponentially. But there is no exponential decay of entanglement entropy.

---

## Round 1 · Author Response

Dear Editor and referees, We are very grateful for the many comments and remarks that helped us increase the quality of our manuscript. We have implemented corrections for all the issues you raised, and added a lot of new content to make the manuscript detailed. Below we list the major changes to the manuscript.

Detailed answers to all issues raised by the referees have been submitted separately as replies.

---

## Round 1 · List of Changes

1. Lines 76-81. We added a description of the structure of the whole paper.
  2. Lines 97-101. We rewrote these sentences to make them clearer.
  3. Lines 107-111. We added descriptions of an output linear layer that maps the final hidden $\hat{h}^{[N]}$ to classifications.
  4. Lines 127-131 & Fig.3. We added visualization and descriptions of t-SNE intermediate data.
  5. Lines 189-191. We added more formulas to state the equivalence of sResMPS and regular MPS.
  6. Lines 222-242. We added Sec. 2.5 to compare ResMPS, RNN, and transformer network.
  7. Appendix A. Detailed training information which helps to reproduce our result.
  8. We uploaded codes to GitHub repository (see YemingMeng/ResMPS).
  9. Appendix B. Comparison of different paths which map 2D figures to 1D sequence.
  10. Appendix C & Tab. 2. Additional benchmarks 1 that compare exhaustive combinations of different components (channels, activation and dropout).
  11. Appendix C. Additional benchmark 2 that compare performance under small virtual feature dimension $\chi<10$.
  12. Several typos were fixed.

---

## Round 2 · Referee Report · Anonymous (Referee 2) · 2023-2-14

Strengths

See previous report.

Report

All my previous concerns regarding presentation and interpretation have been resolved.

---

## Round 2 · Author Response

Dear Editor and referees, we are grateful for your comments and feedback on our manuscript. Accordingly, we have modified the manuscript and mention the changes we have made.

Weaknesses

We thank the referee for pointing out the weaknesses. As the referee mentioned, bound dimension $\chi$ is still valid to measure the representative power of MPS. On the other hand, the pruning test suggests an alternative approach to truncating an MPS, that is, limiting the total number of variational parameters that actually contribute to the prediction accuracy. This leads to a new variant of ResMPS called "sparse ResMPS", whose representative power is characterized by M rather than $\chi$. Based on the above discussion, we have revised section 2.4.3 accordingly.

Requested changes

  1. Please refer to the Weaknesses discussed above.
  2. We concur with the referee's assessment that our argument was lacking in support. We have removed it in the revised version.
  3. We agree with the referee's point of view. TN can be used to represent linear maps, but the essence of TN is more general, which is far beyond what we concluded in this sentence. Therefore, we corrected our statement.
  4. We thank the referee for carefully identifying our problem, we fixed this issue in the revised version.
  5. As the referee's suggestion, we swapped their positions accordingly.
  6. We fixed this issue.
  7. We fixed this issue.

---

## Round 2 · List of Changes

1. Line 48. We corrected the issue in the description of TN as a linear map between quantum states.
  2. Lines 77-78 & Sec. 2.2, Sec. 2.3. We swapped section 2.2 and section 2.3, and updated the structure of the paper paragraph correspondingly.
  3. Line 108. We added the missing "not" to the original text.
  4. Line 116. We corrected the reference to figure 1(f) to correctly refer to figure 1(e).
  5. Line 184. We removed the final sentence that lacked sufficient support.
  6. Sec. 2.4.3. We completely rewrote the subsection.
  7. Lines 361-362. We corrected the description of the entanglement entropy decay of ResMPS.

---

## Editorial Decision

published